# Factors Affecting Preventive Behaviors for Safety and Health at Work during the COVID-19 Pandemic among Thai Construction Workers

**DOI:** 10.3390/healthcare11030426

**Published:** 2023-02-02

**Authors:** Samsiya Khaday, Kai-Way Li, Halimoh Dorloh

**Affiliations:** Department of Industrial Management, Chung Hua University, Hsinchu 300, Taiwan

**Keywords:** COVID-19, integrated protection motivation theory, extended theory of planned behavior, construction worker preventive behavior

## Abstract

Occupational health and safety risks are of major concerns in construction industry. The COVID-19 outbreak provides an additional risk that could drastically affect the safety risks and health of construction workers. Understanding the factors that affect the health and safety of construction workers is significant in reducing risky behaviors and enhancing worker preventive behaviors. Via integrating the Protection Motivation Theory (PMT) and the extended Theory of Planned Behavior (TPB), this study investigates the factors that affect preventive behaviors among construction workers during the COVID-19 pandemic in Bangkok, Thailand. A total of 610 Thai construction workers participated in an online questionnaire survey, which consisted of nine factors with 43 questions. Structure equation modeling (SEM) was adopted to analyze the causal relationships among the latent variable. The SEM results indicated that organizational support and knowledge about COVID-19 had significant (*p* < 0.0001) direct influences on perceived vulnerability and perceived severity. In addition, perceived vulnerability and perceived severity had significant direct influences on perceived behavioral control. Perceived severity had significant (*p* < 0.0001) direct influence on attitude towards behavior. Moreover, perceived behavioral control and attitude towards behavior had significant (*p* < 0.0001) direct influence on intention to follow the preventive measure. Furthermore, the intention to follow the preventive measure had significant (*p* < 0.0001) direct influences on the COVID-19 preventive behavior. Of note, organizational support and knowledge about COVID-19 had significant (*p* < 0.0001) indirect influence on COVID-19 preventive behavior. The findings of this study may assist project managers/supervisors and authorities in the construction industry in understanding the challenge during COVID-19 and possible similar epidemics in the future. In addition, conducting effective strategies would improve construction industry safety and promote preventive behaviors among construction workers.

## 1. Introduction

COVID-19 is caused by the severe acute respiratory syndrome coronavirus 2 (SARS-CoV-2) [1] is a disease affecting all countries in the world. It has been a confirmed as a disease that is transmitted from person to person by close contact. Its main symptoms include fever, dry cough, fatigue, and shortness of breath [2]. Less commonly reported symptoms include sputum production, headache, hemoptysis, and diarrhea [3,4,5,6]. The virus is generally more fatal for the elderly and those with comorbidities, such as hypertension, obesity, diabetes, and kidney disease [7,8,9]. In the face of the pandemic, governments throughout the world mandated and promoted various control measures, such as shutting down public transportation and facilities, testing and tracing affected communities [10], encouraging social distancing between individuals, wearing a facial mask, and hand washing or sanitization [11]. The World Health Organization [11] has also been working with health authorities and pharmaceutical industry globally to develop vaccines for infection testing and medicines for treatment of this disease.

During the pandemic, people displayed the preventive behaviors mentioned by the health authorities to control infection. The literatures have indicated that trust and confidence in health authorities improve compliance [12,13,14]. However, the literature also illustrated that behavior change is a considerable challenge during transmissible disease pandemics [15,16]. Pfattheicher et al. [17] pointed out that empathy is associated with social distancing and wearing a mask. Sympathy with those most at risk from the virus promoted motivation to take preventive measures. Allington et al. [18] reported that conspiracy beliefs and social media usage as a source of information during the COVID-19 public health emergency were related to non-adherence to suggestions for protective behaviors. Furthermore, although there is some initial evidence of the importance of psychological factors, such as fear and information behavior for individual adherence to preventive measures, there is no conclusive evidence.

The COVID-19 pandemic affects all sectors, including the construction sector. It results in work suspensions and new safety protocols [19]. The construction industry is suffering the highest COVID-19 infection rates during the pandemic [20,21]. Construction workers are approximately five times more likely to be hospitalized due to this disease than workers in other sectors [19,22,23]. As COVID-19 dissemination is enormously related to individual close contact, it has dramatically impacted construction workers. Gathering workers is requisite for field jobs on construction sites. Thus, a shift, such as limitations in working duties and working hours [24,25], has occurred with workers who were facing with spreading of the virus. Concurrently, workers are also exposed to physiological and psychological stress [26], fear and anxiety [27], and are susceptible to COVID-19 pandemics [28]. Hence, safety policies and procedures, in addition to the existing ones, have been implemented, including reduced physical interaction, safe distances between people, and new hygiene and personal protective equipment (PPE) measures. These interventions have affected the number of workers permitted to work in each working area.

Understanding the factors affecting the health and safety of construction workers is essential to avoid risky behavior. Lüdecke and Knesebeck [29] indicated that disease-specific knowledge is essential to enable people to judge information on COVID-19 and to reduce the spread of the disease. Dai et al. [30] suggested that systematic intervention programs for governmental factors must be integrated with individual factors to achieve effective prevention and control of COVID-19 among the public. Due to the lack of literature regarding the preventive behavior of Thai construction workers during the COVID-19 pandemic, it is essential to investigate the factors affecting the health and safety-related behavior of construction workers during the COVID-19 pandemic. This study investigates the preventive behavior during the epidemic dispersal of construction workers in Thailand by integrating the Protection Motivation Theory (PMT) [31] and the extended Theory of Planned Behavior (TPB) [32]. The PMT and TPB could holistically approach the behavior and evaluate factors affecting the COVID-19 preventive behavior in the context of construction [33,34].

Different factors were considered in this study, including organizational support during epidemics, knowledge about COVID-19, perceived vulnerability, perceived severity, perceived behavioral control, subjective norm, attitude toward behavior, and intention to follow the preventive measure. These factors were evaluated by utilizing confirmatory factor analysis (CFA) [30] and the structural equation modeling (SEM) approach [31]. The findings are beneficial for the construction industry in supervising the construction workers on their preventive behavior during the epidemic, which is significant for its sustainable disease prevention and management.

## 2. Theoretical Research Framework

PMT refers to humans protecting themselves from the perceived health threat when receiving information about the severity of the risk, the perceived vulnerability, and the opportunity to reduce the risk [34,35,36,37]. Researchers have used PMT as a theoretical framework to understand health-related behaviors and to assess humans’ intentions to engage in preventive behaviors [38,39]. People believe that appropriate preventive behavior may reduce the risk of inaction [40], while TPB refers to the performance of a particular behavior that is determined by the behavioral intention to perform the behavior [41]. It was developed as an instrument by identifying attitudes, self-efficacy, and norms as significant predictors of people’s understanding of their interests in deciding to adopt a changed behavior [37,42].

Considering the current epidemic, reliable and accurate information on current pandemics is essential to distinguish and promote performed prevention efforts by construction workers. As the government promotes measures against the virus, advances in COVID-19 vaccines are possible measures to control the transmission of the virus. However, traditional prevention of non-medical and control methods, such as hand hygiene with alcohol gel and wearing masks, needs to be continued and promoted [43,44]. These essential knowledges about outbreaks can help construction workers avoid public risk behaviors and clarify how they perceive the risk of becoming infected. In light of the above discussions, we have the following hypotheses:

**H1:** *Organizational support had a significant influence on perceived vulnerability*.

**H2:** *Organizational support had a significant influence on perceived severity*.

**H3:** *Knowledge about COVID-19 had a significant influence on perceived vulnerability*.

**H4:** *Knowledge about COVID-19 had a significant influence on perceived severity*.

The PMT has been applied in the research on COVID-19 preventive behaviors [39,45,46,47]. In this study, the TPB was included to determine the significant factors that influence the health behavior of construction workers. There were few studies that address the relationship between PMT and TPB. Prasetyo et al. [33] examined the factors influencing the perceived effectiveness of COVID-19 prevention interventions by integrating PMT and the expanded TPB. Their results showed that perceived vulnerability and perceived severity had a significant indirect effect on intention. The intention had significant direct effects on behavior, which in turn led to perceived effectiveness.

Nguyen et al. [48] integrated PMT and TPB to assess factors influencing the intention to follow COVID-19 prevention interventions. Their results showed that the perceived risk of COVID-19 had a significant impact on people’s attitudes and perceived behavioral control. Trifiletti et al. [49] conducted protective behaviors during the COVID-19 pandemic by using the TPB and risk perception; they recommended that intervention and communication strategies to prevent the spreading of COVID-19 should be strongly organized. Shanka et al. [50] highlighted that awareness of the risk, feelings of responsibility, and moral obligations, influenced compliance behavior. Based on the above discussions, this study correlates the variables of PMT with critical predictors of TPB and proposes the following hypotheses:

**H5:** *Perceived vulnerability had a significant influence on perceived behavioral control*.

**H6:** *Perceived vulnerability had a significant influence on subjective norms*.

**H7:** *Perceived vulnerability had a significant influence on attitudes toward COVID-19*.

**H8:** *Perceived severity had a significant influence on perceived behavioral control*.

**H9:** *Perceived severity had a significant influence on subjective norms*.

**H10:** *Perceived severity had a significant influence on attitudes toward COVID-19*.

Based on TBP, perceived behavioral control refers to individual perception of the difficulties associated with a specific behavior [32]. The perception of how easy or difficult it is to perform a behavior is included [51,52]. As a critical component of behavior change and behavioral prediction, perceived behavioral control promotes positive health intentions and facilitates the persistence of recommended health behaviors [53]. According to Manstead [54], individuals who perceive themselves to have higher behavioral control will exert more effort to accomplish their goals. Based on the PMT variables and critical predictors of TPB, we propose the following hypothesis:

**H11:** *Perceived behavioral control had a significant influence on the intention to follow the preventive measures of COVID-19 pandemics*.

Subjective norms are a person’s beliefs about what others think essentially about someone engaging in a particular behavior and whether they would approve of it [55]. The subjective norm refers to the extent to which construction workers intended to follow the preventive measures of COVID-19 pandemics. The notion of subjective norms plays a critical role in TPB [32]. The TPB refers to subjective norms as perceptions or opinions considered when someone decides to act. When examining the relationship between people’s subjective norms and behavioral intentions, the literature confirms that subjective norms positively affect behavioral intention [41,56,57,58]. When people have more positive subjective norms, they have a stronger intention to act. A hypothesis related to subjective norms was proposed as follows:

**H12:** *Subjective norms had a significant influence on the intention to follow the preventive measures of COVID-19 pandemics*.

Attitudes towards behavior refer to how the perception of the self-performing a particular behavior [33] can be positive or negative. It is commonly assumed that a beneficial result will happen if the desired behavior is practiced [34]. It is evident that attitude is one of the factors affecting behavioral intention. Husain et al. [59] explored the intention to take the COVID-19 vaccine in northern India. Their results showed that attitudes affected the intention of taking the COVID-19 vaccine significantly. However, workplace vaccination campaigns offering on-site vaccination for workers have demonstrated a different success rate [60]. Prasetyo et al. [33] studied the factors affecting the perceived effectiveness of COVID-19 prevention measures. Their results supported the idea that attitude can predict behavioral intention. Based on these studies, the hypothesis was proposed as follows:

**H13:** *Attitudes toward COVID-19 on preventive behaviors had a significant influence on the intention to follow the preventive measures of COVID-19 pandemic*.

An individual’s positive attitude toward a particular behavior strengthens his/her intention to perform the behavior [32]. Mamman et al. [61] stated that intention is an individual’s inclination toward willingness or measurement of motivation towards their plan to perform a behavior [34]. It is the best possible predictor of an individual’s action, despite the many factors affecting preventive behaviors. Intention to follow preventive behavior is crucial during the pandemic. Norman et al. [62] pointed out that individuals’ intentions to interact in action predicted later compliance with protective behaviors, such as avoiding visiting relatives or friends. Based on these studies, the hypothesis was proposed as follows:

**H14:** *Intention to follow the preventive measures of COVID-19 pandemic had a significant influence on the preventive behaviors of COVID-19 pandemic*.

Figure 1 shows the theoretical research framework of the study to explain factors affecting the health and safety of construction workers. Integrating TPB and PMT leads to 14 hypotheses that investigate the relationship between determined and latent variables.

## 3. Methodology

### 3.1. Participation and Sampling Design

A questionnaire survey was administered for construction workers in Bangkok, Thailand. Due to the COVID-19 pandemic, the questionnaire was distributed through an online survey. The study was focused on workers who work in field jobs, including field engineers, construction inspectors, construction forepersons, general laborers, etc. The entire participants were in divergent projects. A manager or representative of each construction project was inquired and contributed an online questionnaire to their workers and co-workers. In addition, construction workers who are able to read in Thai were requested. The data collection took place between 8 August and 3 October 2022. A convenience sample of 610 construction workers was collected to distribute the online questionnaire. Cochran’s formula was adopted to calculate the minimum sample size with a confidence level of 96%, ±5% precision. Table 1 shows the demographic data of the participants. Most participants were between 25 and 34 years old (38.0%). Most of them were male (62.5%). The primary education levels were middle high school (32.6%) and primary school (23.1%). In addition, over 70% of participants were vaccinated by at least one shot for COVID-19 protection. More than half of the participants had COVID-19 (66.2%) once or more.

### 3.2. Questionnaire

From the conceptual framework, we developed a questionnaire to identify the factors that affect health and safety during the COVID-19 pandemic and the ongoing preventive behavior efforts of construction workers. An online questionnaire was created and distributed. The online questionnaire entails ten sections, including demographics, organizational support, knowledge about COVID-19, perceived vulnerability, perceived severity, perceived behavioral control, subjective norm, attitude toward behavior, and intention to follow and COVID-19 preventive behavior. The 43 items were utilized based on the supporting references (see Table 2). The constructs in each item were measured using a 5-point Likert scale from “strongly disagree (1)” to “strongly agree (5)”.

The item measurements were identified through both literature review and a focused group panel discussion with six professionals in construction safety, including three construction managers, a safety consultant, a professor of civil engineering, and a professor of industrial engineering. The professors reviewed language appropriation. The managers reviewed the fitness of items measurement. Meanwhile, the consultant reviewed the propriety of the questionnaire. Content validity was used to assess how well the instrument was designed. The professionals were asked to rate each of 43 items, with the result being that all 43 items having a content validity index of 1.0. Therefore, all items demonstrated excellent content validity [63]. Subsequently, a pretest questionnaire was conducted to affirm the comprehensibility of the questionnaire for construction workers.

**Table 2 healthcare-11-00426-t002:** Constructs of measurement items.

Constructs	Items	Measures	References
Organizational support	OS1	I receive organizational support during COVID-19 pandemic.	[59,64,65]
	OS2	My company provides sufficient personal protective equipment for the workers (i.e., gloves, facemasks, alcohol, face shields, and hand washing).	
	OS3	When the COVID-19 pandemic broke out, the company urgently established a pandemic prevention committee.	
	OS4	Management promotes internal communication on COVID-19 prevention via newsletter, e-mail, Facebook, etc.	
	OS5	My managers always try to enforce safety rules and procedures for COVID-19 prevention at the workplace.	
Knowledge about COVID-19	KN1	I do understand the transmission of COVID-19.	[66,67,68,69]
	KN2	I do understand the incubation period of COVID-19.	
	KN3	I do understand the symptoms of COVID-19.	
	KN4	I do understand the protocol if I have symptoms that might lead to COVID-19.	
	KN5	I do understand which hospital in local can treat COVID-19 patients.	
Perceived vulnerability	PV1	I live in an environment where I can be exposed to COVID-19 infection.	[70,71,72]
	PV2	I think I will be infected with COVID-19 easier than others.	
	PV3	I think my friends/colleagues are vulnerable to COVID-19.	
	PV4	I think Thailand is more vulnerable than other ASEAN countries.	
	PV5	I think that I could become infected with COVID-19 through vaccination.	
Perceived severity	PS1	I believe that COVID-19 is highly dangerous.	[73,74]
	PS2	I find COVID-19 may lead to sudden death.	
	PS3	I find COVID-19 may affect my mental health.	
	PS4	I think COVID-19 in Thailand is more severe than in other ASEAN countries.	
	PS5	If my family member were infected with COVID-19, I would keep it a secret.	
Perceived behavioral control	PBC1	The preventive protocols are completely up to me.	[33,56]
	PBC2	I think preventive protocols are easy to be implemented.	
	PBC3	I think although I am healthy, I still have a chance to spread COVID-19 to others.	
	PBC4	I think the risk of death caused by COVID-19 infection is great.	
	PBC5	I think COVID-19 is highly dangerous.	
Subjective Norm	SN1	Most people I know are following the preventive protocols given by the government.	[33,56,75,76]
	SN2	Most people I know are staying at home and work from home.	
	SN3	Most people I know are using hand sanitizer.	
	SN4	Most people I know, are keeping physical distancing.	
Attitude toward behavior	AB1	I feel insecure if someone stand too close to me during the COVID-19 outbreak.	[77,78,79,80]
	AB2	I am worried about myself, my family members, and colleagues who may be affected by COVID-19.	
	AB3	I am scared of individuals coming from the affected areas.	
	AB4	I worry about the number of people infected by COVID-19.	
	AB5	I feel stressed during the COVID-19 outbreak.	
Intention to follow	IF1	I intend to follow the recommended precautions until the end of the COVID-19 outbreak.	[2,81]
	IF2	I intend to follow every rule by the government during the COVID-19 pandemic.	
	IF3	I intend to continue to use standard control measures.	
	IF4	I intend to follow my company’s protocol during the spreading of COVID-19.	
COVID-19 preventive behavior	PB1	I usually wear a facial mask when I leave home.	[29,30,82,83]
	PB2	I embrace personal hygiene practices and washed my hands more often and longer.	
	PB3	I wash my hands or clean them with alcohol often.	
	PB4	I usually wear gloves when interacting with shelf materials/products.	
	PB5	I practice 1-meter social distancing to reduce unnecessary infection.	

### 3.3. Reliability and Validity Assessments

The internal consistency reliability for the COVID-19 preventive behavior was assessed using Cronbach’s α. The SPSS^®^ 20 (IBM^®^, Armonk, NY, USA) was adopted to test the reliability of each factor.

The validity of the model was tested for convergence and discriminate validity [84]. To assess convergent validity, each factor loading and average variance extracted (AVE) were assessed. According to the literature [84,85,86,87,88], the factor loading and AVE values must exceed 0.70 and 0.50, respectively. In addition, to test the discrimination validity of each construct of organizational support, knowledge about COVID-19, perceived severity, perceived vulnerability, perceived behavioral control, subjective norm, attitude toward behavior, intention to follow, and preventive behavior, and correlations between the AVE square roots values of each construct were computed.

### 3.4. Structural Equation Modeling

To verify which measured variable is related to which latent variable, confirmation factor analysis (CFA) was used. We utilized structural equation modeling (SEM) to evaluate how the TPB and PMT conceptual theories interact for COVID-19 preventive behaviors [33,41,89,90,91]. The hypotheses were tested using the SPSS^®^ 20 and AMOS 22. Several fitness measures were applied to confirm the validity of the model [89,92]. These measures include incremental fitness index (IFI), Tucker Lewis index (TLI), comparative fit index (CFI), and root mean square error of approximation (RMSEA).

## 4. Result

CFA was conducted to verify the validity and relationship of the variables used to measure the study variable. Seven items (KN4, PV5, PS5, SN1, SN2, SN3, and SN4) with a factor loading less than 0.5 were removed [35]. Therefore, the subjective norm was removed from the model, 36 items with eight indicators were retained on the scale. In addition, IFI, TLI, and CFI were adopted as important indicators to confirm the validity of the model, indicating that all model fitness was considered significantly acceptable with a value greater than 0.90 [89,92]. Furthermore, RMSEA with a value of less than 0.08 is considered acceptable (Table 3) [89].

The SEM modification indices were applied to improve the model’s fit by evaluating the COVID-19 preventive behavior of Thai construction workers. As shown in Figure 2, the final SEM assesses COVID-19 preventive behavior among workers. The model determined that the following hypotheses were insignificant: perceived vulnerability to attitude toward behavior (hypothesis 6), perceived vulnerability to the subjective norm (hypothesis 7), perceived severity to perceived behavioral control (hypothesis 8), perceived vulnerability to subjective norms (hypothesis 9), and subjective norms to intentions to follow (hypothesis 12). Therefore, we removed the insignificant hypotheses to improve the model. Figure 3 shows the final model obtained from the COVID-19 preventive behavior measure. Table 4 shows the initial and final factor loading of each indicator from the SEM for evaluating preventive behavior during the COVID-19 pandemic. Moreover, the overall model fitness indices indicated that IFI, TFI, CFI, and RMSEA were all acceptable (Table 5).

Table 6 illustrates the reliability and validity test results of this study. The composite reliability values were more significant than 0.70, demonstrating that the constructs were valid and had overall reliability [33,93]. In addition, all the Cronbach α values for the internal consistency reliability testing were higher than 0.7, indicating that the reinforced constructs were reliable and appropriate for each item on the specified latent construct. In terms of the average variance extracted (AVE), the AVE of the nine indicators were all higher than 0.50, indicating that they have excellent convergence and suitable convergent validity.

The square root of AVE was between 0.79–0.94, and these values were higher than the correlations among constructs, indicating suitable discriminant validity (see Table 7). Table 8 shows the effects (direct, indirect, and total) between the causal relationship of the latent variable. The *p*-values for all the paths of direct and indirect effects were less than 0.05. The intention to follow COVID-19 preventive behavior was found to be the highest direct effect, while perceived behavioral control of COVID-19 preventive behavior was found to be the highest indirect effect.

## 5. Discussion

This study integrated PMT and Extended TPB in analyzing the preventive behavior among construction workers during the COVID-19 epidemic dispersal in Bangkok, Thailand. Construction workers generally have a low education. Most of construction workers in our study (64.4%) had educations that were lower than the high school level. The majority of them were male (62.5%). The primary age group (38%) was between 25 and 34 years old. Moreover, 2.8% of the workers were below 15 years old. In Thailand, the Labour Protection Act (chapter 5, Clause 45, B.E 2541, 1998) requires that an employer shall not employ a child under fifteen years of age unless they have graduated from a middle school or the competent authority has determined that the work does not cause any physical and mental harm to the young workers [94]. Since the COVID-19 outbreak, the construction camp has been ordered to stop, leading to a post-epidemic labor shortage. The young workers who joined our study were approved by local authority to work on construction sites as helpers to senior worker in painting, carpentry, and cement works.

The study distributed an online questionnaire containing 43 questions regarding the preventive behavior among construction workers during epidemic dispersal. Subjective norm (H6, H7, H8, H9, H12) was not accepted by the PMT and Extended TPB model. In addition, SEM was utilized to investigate the interrelationship among the latent variables. Among these variables, the reliability assessment results indicated acceptance of the internal consistency reliability of COVID-19 preventive behavior. The validity assessment results showed that the convergent and discriminant validities of construction workers’ preventive behaviors were all acceptable. The SEM results revealed direct and indirect relationships that affected the preventive behaviors among construction workers during COVID-19 epidemic dispersal.

### 5.1. Theoretical Implications

From the perspective of Thai construction workers, the integrated PMT and Extended TPB in this study confirmed that organizational support of construction workers during epidemics had a significant influence on PV (β = 0.404; *p* < 0.0001; OS→PV; H1) and PS (β = 0.349; *p* < 0.0001; OS→PS; H2). The organizational theory proposes that employees form a generalized perception concerning the extent to which the organization values their contributions and cares about their well-being [95,96]. Due to the ongoing epidemic, organizational support is significant for workers to reduce the dispersal of the COVID-19: for instance, providing sufficient personal protective equipment for the workers, enforcing safety rules and procedures at the workplace, and providing appropriate communication equipment. Hence, the organization could appreciate the negative effect of the virus by implementing support measures. However, the seriousness of COVID-19 preventive measures is needed to enhance the perceived vulnerability and perceived severity of the virus disease.

Regarding the knowledge of COVID-19, the SEM revealed that KN has a significant influence on PV (β = 0.378; *p* < 0.0001; KN→PV; H3) and PS (β = 0.319; *p* < 0.0001; KN→PS; H4). Knowledge of COVID-19 related to understanding the transmission and incubation period of COVID-19 disease, viral protocol symptoms that could lead to COVID-19 disease, and how hospitals treating COVID-19 patients, would positively influence perceived vulnerability and severity. These are essential for the preventive spread of the virus. Prasetyo et al. [33] indicated that an understanding of COVID-19 among Filipinos during reinforced community quarantine significantly influences perceived vulnerability and severity. Thus, if workers receive more accurate COVID-19 information, they could better understand the disease and its effects and symptoms. This could increase their perceived vulnerability and severity.

In terms of perceived vulnerability, the results showed that PV has a significant influence on PBC (β = 0.426; *p* < 0.0001; PV→PBC; H5). This suggests that increased perceived vulnerability to epidemic disease leads to increased susceptibility to epidemic disease cues, interpersonal avoidance, and extreme attitudes that may reduce engagement with others, especially when the potential threat of infection becomes apparent [91,92,93,94,95,96,97,98,99,100]. The results show that perceived vulnerability to disease corresponds with preventive behaviors and may promote transformation to COVID-19 pandemic.

The behavioral perspective of construction workers towards the COVID-19 pandemic is affected when they are living in an environment which may be exposed to COVID-19 infection. They may be infected with the disease. Their friends/colleagues may be vulnerable to COVID-19 and they believe they could become infected with the virus through vaccination. The PBC indicators emphasized that workers are vulnerable to disease, its effects, and its symptoms. Protocols for infection prevention and control of the spreading of the COVID-19 disease measure are needed. On the other hand, perceived severity significantly influenced AB (β = 0.430; *p* < 0.0001; PS→AB; H10). In addition, the behavioral perspective of construction workers towards the COVID-19 pandemic is affected due to mental health effects. Workers who perceive the severity of COVID-19 disease, its effect, and its symptoms are more concerned with its mental health effects (such as worry, stress, and fear about being infected). This is consistent with the findings of Prasetyo et al. [33].

Regarding perceived behavioral control, the results indicated that PBC had a significant influence on AB (β = 0.348; *p* < 0.0001; PBC→IF; H11). This implies that construction workers’ willingness to follow government protocol guidance and support of preventive measures helps in preventing the spread of the disease. Furthermore, attitude toward behavior (AB) was found to significantly influence IF (β = 0.348; *p* < 0.0001; AB→IF; H13). The indicators such as security, scare, anxiousness, and stress considerably affect the intention to follow preventive measure during the spreading of the virus. In addition, intention to follow (IF) was found significant in influencing PB (β = 0.538; *p* < 0.0001; IF→PB; H14). This implies that construction workers’ willingness to follow is likely to enhance preventive behavior measures during the spreading of the current disease toward the recommended precautions. Moreover, IF is likely to affect some precautions, such as wearing a face mask, embracing personal hygiene practices, washing hands, using alcohol, and avoiding crowded places that would affect their infection of COVID-19 disease.

Knowledge about COVID-19 had a significant indirect effect on COVID-19 preventive behavior (β = 0.195; *p* < 0.0001). Providing knowledge to workers leads to a promotion of construction workers’ preventive behaviors, during the COVID-19 pandemic. This knowledge includes transmission of COVID-19, its incubation period, its symptoms and effects, the protocol if one has COVID-19 symptoms, and hospitals that can treat COVID-19 patients. Thus, management should strive to provide this knowledge to workers. In addition, organizational support had a significant indirect effect on COVID-19 preventive behavior (β = 0.295; *p* < 0.0001). Establishing a pandemic prevention committee, promoting internal communication on COVID-19 prevention via media, and specifying safety rules and procedures for COVID-19 prevention at the workplace would significantly impede the spreading of the virus. The perceived severity had a significant indirect effect on COVID-19 preventive behavior (β = 0.338; *p* < 0.0001). People who perceive a higher severity of the disease are more likely to take precautionary actions recommended by public health authorities. This was consistent with the findings of Park and Oh [71] and Luo et al. [101].

### 5.2. Theoretical Contribution

This study enhances our knowledge of the preventive behavior during the COVID-19 pandemic of construction workers in Thailand. The model integrated the theory of planned behavior and the protection motivation theory, which provide new insight into construction workers’ preventive behaviors during COVID-19 pandemic in Thailand. Through the literature review and the findings of this study, we identified organizational support toward COVID-19 pandemic and knowledge about COVID-19 as significant variables affecting the preventive behavior of construction workers during the epidemic. In addition, we utilized structural equation modeling to determine the relationship between the latent variables.

### 5.3. Practical Implications

Our empirical findings have important policy implications for COVID-19 preventive behavior. The integration of the TPB and the PMT are essential for assessing the relationship among latent variables on the COVID-19 preventive behavior of construction workers. The TPB and PMT model would lead to developing guidelines for workers to make appropriate preventive behavior during COVID-19 pandemic. Developing further prevention measures and strategies for managing the spreading of the virus in the construction industry is needed to reduce the risk of virus infection. In addition, workers with high perceived organizational support are more likely to take preventive behavior during COVID-19 pandemic.

Meanwhile, workers with more knowledge and understanding of COVID-19 have significantly higher motivation to take preventive behavior. The evidence of this study indicates that the relationship among seven indicators (organizational support, knowledge about COVID-19, perceived vulnerability, perceived severity, perceived behavioral control, attitude toward behavior, and intention to follow) had significant effects on construction workers’ preventive behaviors during COVID-19 pandemic. This could be implemented as a preventative measure for construction workers in the industry to avoid risky behavior during the COVID-19 pandemic.

In addition, the SARS-CoV-2 vaccine is significant in minimizing the infection risks and reducing the severity of infection and hospitalizations. Occupational physicians play a crucial role in implementing vaccination. Vaccination information for workers is needed to provide them with a proper vaccine. Occupational physicians’ workplace vaccine campaigns should be organized to provide thorough vaccinations to workers to prevent the spread of the virus and promote workplace safety. Occupational physicians have to work under the risk of COVID-19 infection. The practice to protect those physicians should not be neglected.

### 5.4. Limitations

There are limitations of this study. First, this study was based on the variables of the TPB and PMT. It is irresolute whether these individual factors can always prevent COVID-19 infection. Regarding the tool for assessing subjective norms, factor loading of all items below 0.5 and low explanatory power were removed [102]. Thus, the subjective norm was not included in measuring SEM. Moreover, the data were collected via an online questionnaire. Further research to collect more samples among construction workers and administering a face-to-face questionnaire survey may lead to more comprehensive results. Finally, skilled and non-skilled workers may have different preventive behaviors on construction sites. Unfortunately, we could not compare and discuss the difference between these two types of workers, because we did not differentiate the workers into these two types on the survey. This may be an interesting topic in future research.

## 6. Conclusions

The COVID-19 pandemic is a global crisis. Construction workers are increasingly infected with this disease. To prevent the transmission of COVID-19 among workers as a whole, this study provides a thorough investigation on implementing TPB and PMT to evaluate the factors affecting preventive behavior during COVID-19 among Thai construction workers. The results of the structural equation model (SEM) indicated that organizational support and knowledge about COVID-19 significantly influenced the perceived vulnerability and perceived severity. In addition, perceived vulnerability and perceived severity had significant direct influences on perceived behavioral control. Perceived severity had significant direct influence on attitude towards behavior. Additionally, perceived behavioral control and attitude towards behavior had significant direct influence on intention to follow the preventive measure. Furthermore, the intention to follow the preventive measure had a significant influence on COVID-19 preventive behavior. Surprisingly, organizational support and knowledge about COVID-19 had significant indirect influence on COVID-19 preventive behavior. To promote construction workers preventive behavior during the COVID-19 pandemic, managers/supervisors and authorities need to educate workers on the evidence of COVID-19 preventive behavior so as to enhance their understanding and active implementation of the best behaviors.

## Figures and Tables

**Figure 1 healthcare-11-00426-f001:**
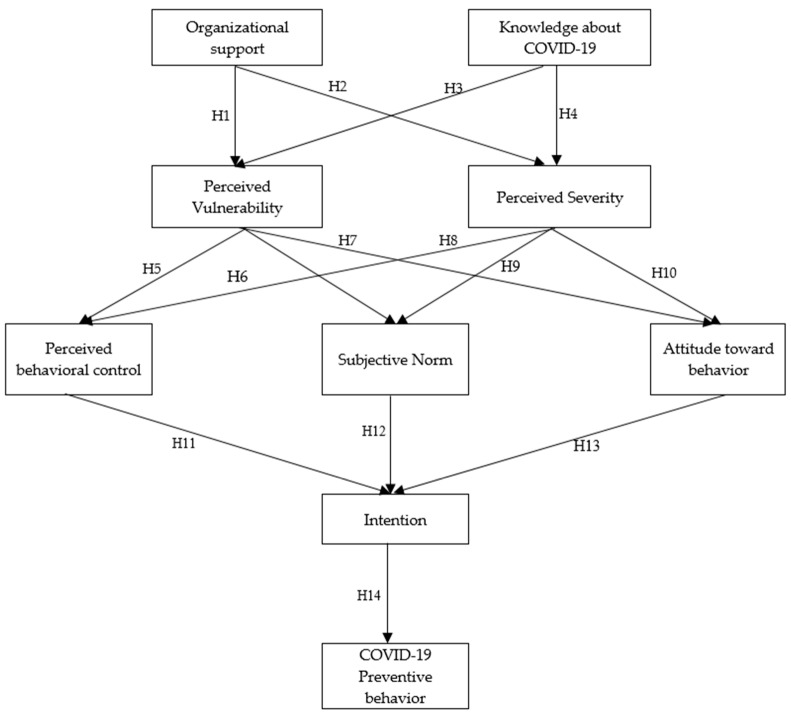
Protective behaviors of construction workers during COVID-19 pandemic model.

**Figure 2 healthcare-11-00426-f002:**
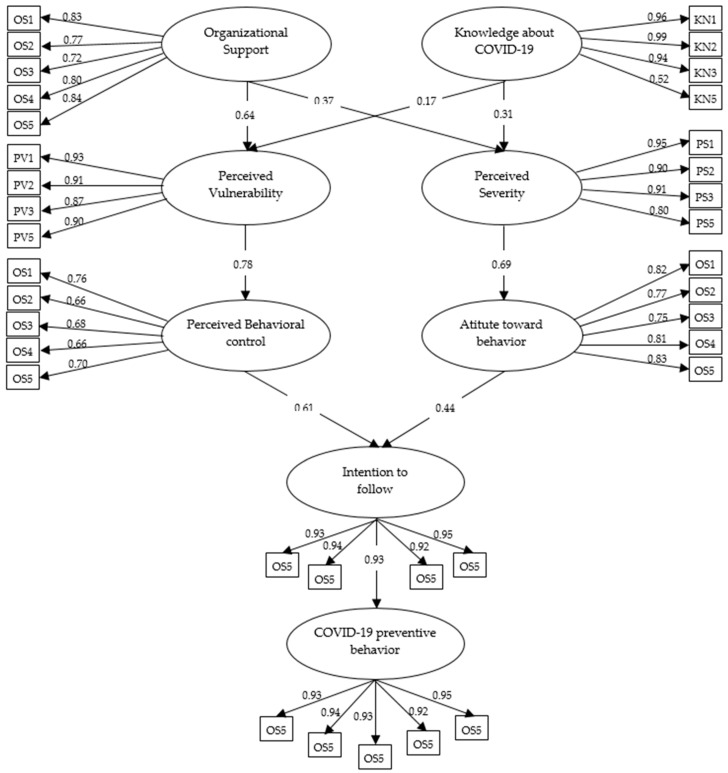
Final SEM with an indicator for assessing the COVID-19 preventive behavior measure.

**Figure 3 healthcare-11-00426-f003:**
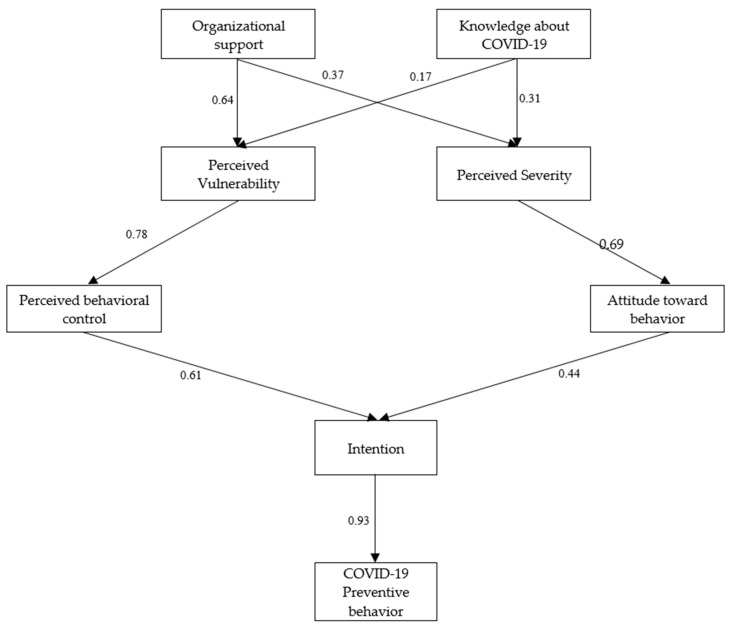
The final model obtained from the COVID-19 preventive behavior measure.

**Table 1 healthcare-11-00426-t001:** Descriptive statistics of respondent (*n* = 610).

Characteristics	Category	N	%
Gender	Male Female	381 229	62.5 37.5
Age group	Below 15 years old 15–24 years old 25–34 years old 35–44 years old 45–54 years old	17 55 232 185 94	2.8 9.0 38.0 30.0 15.4
	55–64 years old Above 64 years old	27 -	4.4 -
Educational level	No formal education Primary school Middle school High school Post-secondary	53 141 199 106 111	8.7 23.1 32.6 17.4 18.2
Vaccinated COVID-19 protection	Yes No	456 154	74.8 25.2
Detected COVID-19 disease	Yes No	404 206	66.2 33.8

**Table 3 healthcare-11-00426-t003:** CFA Model fitness indices.

Goodness of Fit Measures	Parameter Estimates	Minimum Cutoff
Incremental Fitness Index (IFI)	0.99	>0.90 ^a^
Tucker Lewis Index (TLI)	0.99	>0.90 ^b^
Comparative Fit Index (CFI)	0.98	>0.90 ^a^
Root Mean Square Error of Approximation (RMSEA)	0.04	<0.08 ^a^

^a^ [89]; ^b^ [92].

**Table 4 healthcare-11-00426-t004:** Descriptive statistic results.

Factors	Items	Mean	STD	Factor Loading	
				Initial	Final
Organizational support	OS1	4.09	0.75	0.81	0.83
	OS2	4.07	0.79	0.80	0.77
	OS3	4.04	0.81	0.75	0.72
	OS4	4.11	0.74	0.80	0.80
	OS5	4.16	0.66	0.80	0.84
Knowledge about COVID-19	KN1	4.27	0.63	0.89	0.96
	KN2	4.29	0.64	0.91	0.99
	KN3	4.22	0.68	0.89	0.94
	KN5	3.95	1.06	0.62	0.52
Perceived Vulnerability	PV1	4.12	0.66	0.85	0.93
	PV2	4.05	0.73	0.85	0.91
	PV3	4.01	0.74	0.84	0.87
	PV4	4.07	0.72	0.83	0.90
Perceived Severity	PS1	4.22	0.65	0.81	0.95
	PS2	4.25	0.65	0.83	0.90
	PS3	4.28	0.66	0.83	0.91
	PS4	4.24	0.65	0.82	0.80
Perceived behavioral control	PBC1	4.14	0.69	0.75	0.76
	PBC2	4.10	0.73	0.69	0.66
	PBC3	4.09	0.71	0.69	0.68
	PBC4	4.04	0.72	0.66	0.66
	PBC5	4.08	0.70	0.76	0.70
Attitude toward behavior	AB1	4.04	0.81	0.74	0.83
	AB2	3.89	0.81	0.75	0.81
	AB3	3.91	0.81	0.75	0.75
	AB4	3.95	0.81	0.76	0.77
	AB5	4.04	0.76	0.71	0.82
Intention to follow	IF1	4.40	0.60	0.88	0.93
	IF2	4.46	0.60	0.89	0.94
	IF3	4.51	0.57	0.90	0.92
	IF4	4.42	0.59	0.89	0.95
COVID-19 Preventive behavior	PB1	4.35	0.59	0.70	0.80
	PB2	4.34	0.63	0.87	0.95
	PB3	4.36	0.58	0.87	0.94
	PB4	4.32	0.62	0.87	0.99
	PB5	4.34	0.63	0.87	0.97

**Table 5 healthcare-11-00426-t005:** SEM Model fitness indices.

Goodness of Fit Measures	Parameter Estimates	Minimum Cutoff
Incremental Fit Index (IFI)	0.91	>0.90 ^a^
Tucker Lewis Index (TLI)	0.90	>0.90 ^b^
Comparative Fit Index (CFI)	0.91	>0.90 ^a^
Root Mean Square Error of Approximation (RMSEA)	0.70	<0.08 ^a^

^a^ [89]; ^b^ [92].

**Table 6 healthcare-11-00426-t006:** Reliability and validity assessment.

Factors	Cronbach’s Alpha	Composite Reliability	AVE
Organizational support	0.89	0.79	0.63
Knowledge about COVID-19	0.87	0.87	0.76
Perceived Vulnerability	0.95	0.89	0.80
Perceived Severity	0.69	0.71	0.52
Perceived behavioral control	0.94	0.80	0.63
Attitude toward behavior	0.90	0.94	0.87
Intention to follow	0.97	0.86	0.74
COVID-19 Preventive behavior	0.97	0.93	0.87

**Table 7 healthcare-11-00426-t007:** Correlations of latent variables and verification of construct validity.

Variables	OS	KN	PV	PS	PBC	AT	IF	PB
OS	0.79							
KN	0.22 **	0.87						
PV	0.31 **	0.33 **	0.89					
PS	0.28 **	0.29 **	0.40 **	0.71				
PBC	0.20 **	0.19 **	0.30 **	0.32 **	0.80			
AT	0.26 **	0.20 **	0.29 **	0.40 **	0.2 **	0.94		
IF	0.15 **	0.20 **	0.18 **	0.32 **	0.17 **	0.31 **	0.86	
PB	0.21 **	0.28 **	0.32 **	0.36 **	0.29 **	0.47 **	0.50 **	0.93

Note: The diagonal values are square root of average variance extracted values (AVE), ** *p* < 0.01.

**Table 8 healthcare-11-00426-t008:** Direct, indirect, and total effects.

Variable	Direct Effect	*p*-Value	Indirect Effect	*p*-Value	Total Effect	*p*-Value
OS→PV	0.404	<0.0001	-	-	0.404	<0.0001
OS→PS	0.349	<0.0001	-	-	0.349	<0.0001
OS→AB	-	-	0.330	<0.0001	0.330	<0.0001
OS→PBC	-	-	0.342	<0.0001	0.342	<0.0001
OS→IF	-	-	0.137	0.001	0.137	0.001
OS→PB	-	-	0.295	<0.0001	0.295	<0.0001
KN→PV	0.378	<0.0001	-	-	0.378	<0.0001
KN→PS	0.319	<0.0001	-	-	0.319	<0.0001
KN→AB	-	-	0.164	<0.0001	0.164	<0.0001
KN→PBC	-	-	0.151	<0.0001	0.151	<0.0001
KN→IF	-	-	0.114	0.002	0.114	0.002
KN→PB	-	-	0.195	<0.0001	0.195	<0.0001
PV→PBC	0.426	<0.0001	-	-	0.426	<0.0001
PV→IF	-	-	0.127	0.003	0.127	0.003
PV→PB	-	-	0.246	<0.0001	0.246	<0.0001
PS→AB	0.430	<0.0001	-	-	0.430	<0.0001
PS→IF	-	-	0.263	<0.0001	0.263	<0.0001
PS→PB	-	-	0.438	<0.0001	0.438	<0.0001
PBC→IF	0.183	<0.0001	-	-	0.183	<0.0001
PBC→PB	-	-	0.481	<0.0001	0.481	<0.0001
AB→IF	0.348	<0.0001	-	-	0.348	<0.0001
AB→PB	-	-	0.370	<0.0001	0.370	<0.0001
IF→PB	0.538	<0.0001	-	-	0.538	<0.0001

## Data Availability

Data are available upon request.

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
