# Peer review of "Factors Affecting Preventive Behaviors for Safety and Health at Work during the COVID-19 Pandemic among Thai Construction Workers"

_healthcare, 2023, doi:10.3390/healthcare11030426_

Round 1
Reviewer 1 Report (Previous Reviewer 3)
The manuscript entitled “Factors Affecting Preventive Behaviors for Safety and Health at Work During the COVID-19 Pandemic Among Thai Construction Workers” has no major problems, but there are still some details need to be optimized..
The design of the questionnaire is actually not rigorous enough. A simple reliability test does not ensure the appropriateness of the questionnaire dimensions and questions to the study content. Perhaps the questionnaire design could be described in more detail or added to the limitations.
Author Response
The manuscript entitled “Factors Affecting Preventive Behaviors for Safety and Health at Work During the COVID-19 Pandemic Thai Construction Workers” has no major problems, but there are still some details need to be optimized.
The design of the questionnaire is actually not rigorous enough. A simple reliability test does not ensure the appropriateness of the questionnaire dimensions and questions to the study content. Perhaps the questionnaire design could be described in more detail or added to the limitations.
Response: Thank you for your comment. The design of the questionnaire was added on lines 228-238.
The item measurements were identified through the literature review (e. g., COVIS-19 information, COVID-19 safety guidance, COVID-19 preventive behavior, and infection of COVID-19 statistics). We focus group discussion of six professionals in construction safety, including three construction managers, a safety consultant, a professor of civil engineering, and a professor of industrial engineering. The professors reviewed the language appropriation. The managers reviewed the fitness of items measurement. Meanwhile, the consultant reviewed the propriety of the questionnaire. Content validity was used to assess the instrument. The professionals were asked to rate each of 43 items, resulting that all 43 items had a content validity index of 1.0. Therefore, all items demonstrated excellent content validity (Polit & Beck, 2006).
Reviewer 2 Report (Previous Reviewer 1)
This study investigates the factors affecting the health and safety of construction workers to avoid at-risk behaviours.
Introduction is appropriate, briefly framing the OCVID-19 pandemic and its effect in workers and specifically in construction workers. It also explains why the authors decided to investigate the selected parameters: organizational support during epidemics, knowledge about COVID-19, perceived vulnerability, perceived severity, perceived behavioural control, subjective norm, attitude toward behaviour, and intention to follow preventive measures.
The theoretical framework has been thoroughly explained to the reader, making the research hypotheses understandable to the reader. In this paragraph, the COVID-19 vaccination uptake in workers is mentioned, which would be a very interesting approach, but could be detailed more in depth since workplace vaccination campaigns have shown varying success rates (just as an example: Gualano et al. “Employee participation in workplace vaccination campaigns: a systematic review and meta-analysis”).
Materials and method are presented well, and the use of an online questionnaire survey, which consisted of 9 factors with 43 questions, is quite practical and has allowed for a valid sample. Structure equation model (SEM) was adopted to analyse the causal relationships 17 among the latent variable construct.
Results are well detailed, with integration of graphs and tables that makes them easier to understand to the reader. Framework appears clear and well explained.
The discussion is comprehensive and takes up what has been published in the previous literature on this topic. Considering SARS-CoV2 infection and vaccine-preventable diseases, I suggest to highlight the importance of occupational physicians in implementing workplace campaigns, as discussed in the introduction, framing the importance they have but also the medical liability problems emerged for the COVID-19 vaccination administration (for example, see Beccia et al: “COVID-19 Vaccination and Medical Liability: An International Perspective in 18 Countries”).
Conclusions are complete and encompass all the points discussed above in a clear way. Despite the complexity of the topic the authors explain thoroughly the framework, the hypotheses and methodology, adding to the existing literature with their results.
Author Response
This study investigates the factors affecting the health and safety of construction workers to avoid at-risk behaviours.
Introduction is appropriate, briefly framing the OCVID-19 pandemic and its effect in workers and specifically in construction workers. It also explains why the authors decided to investigate the selected parameters: organizational support during epidemics, knowledge about COVID-19, perceived vulnerability, perceived severity, perceived behavioural control, subjective norm, attitude toward behaviour, and intention to follow preventive measures.
Response: Thank you for your comment.
The theoretical framework has been thoroughly explained to the reader, making the research hypotheses understandable to the reader. In this paragraph, the COVID-19 vaccination uptake in workers is mentioned, which would be a very interesting approach, but could be detailed more in depth since workplace vaccination campaigns have shown varying success rates (just as an example: Gualano et al. “Employee participation in workplace vaccination campaigns: a systematic review and meta-analysis”).
Response: Thank you for your comment. The sentence was added on lines 176-177
Workplace vaccination campaigns by offering on-site vaccination for workers have demonstrated a different success rate.
Materials and method are presented well, and the use of an online questionnaire survey, which consisted of 9 factors with 43 questions, is quite practical and has allowed for a valid sample. Structure equation model (SEM) was adopted to analyse the causal relationships 17 among the latent variable construct.
Response: Thank you for your comment.
Results are well detailed, with integration of graphs and tables that makes them easier to understand to the reader. Framework appears clear and well explained.
Response: Thank you for your comment.
The discussion is comprehensive and takes up what has been published in the previous literature on this topic. Considering SARS-CoV2 infection and vaccine-preventable diseases, I suggest to highlight the importance of occupational physicians in implementing workplace campaigns, as discussed in the introduction, framing the importance they have but also the medical liability problems emerged for the COVID-19 vaccination administration (for example, see Beccia et al: “COVID-19 Vaccination and Medical Liability: An International Perspective in 18 Countries”).
Response: Thank you for your comment. The sentences were added on lines 454-462
In addition, the SARS-CoV-2 vaccine is significant in minimizing the infection risk, reducing the severity of infection and hospitalizations. Occupational physicians play a crucial role in implementing vaccination. Vaccination information for workers is needed to provide them with a proper vaccine. Occupational physicians’ workplace vaccine campaigns should be organized to provide thorough vaccination to workers to prevent the spreading of the virus and promote workplace safety. Highlight, Occupational physicians have to work under conditions of uncertainty during COVID-19 pandemic. The policy to protect occupational physicians who administer a vaccination during COVID-19 pandemic could not be neglected.
Conclusions are complete and encompass all the points discussed above in a clear way. Despite the complexity of the topic the authors explain thoroughly the framework, the hypotheses and methodology, adding to the existing literature with their results.
Response: Thank you for your comment.
Reviewer 3 Report (New Reviewer)
Dear Authors,
Thank you so much for providing this paper on "Factors affecting preventive behaviors for safety and health at work during the Covid-19 pandemic among Thai construction workers". The study adopted the Ajzen's TPB and the Protection Motivation Theory to explore the determinants of preventive behaviors for safety and health at work during the Covid-19 pandemic. The topic of this study is very interesting. However, the following points are worthy of consideration to improve the manuscript’s comprehensibility.
Introduction
1. The study's motivation needs to be clearly demonstrated.
Theoretical Research Framework
2. As you have mobilized the Ajzen's TPB and the Protection Motivation Theory to develop your theoretical model, I suggest that you provide a description of these two theories before proceeding with the development of the hypotheses.
3. Figure 1 illustrates your model, which is generally resulting from a theoretical construction; I suggest placing it at the end of the theoretical research framework section.
4. The Ajzen's TPB suggests considering the relationships between the three determinants of behavioral intention, i.e., subjective norms, attitudes, and PBC. I am curious about the reason behind your discarding these hypotheses.
5. I suggest that you revise the research hypothesis phrasing. Please change "a significant influent "by using "a significant influence ".
6. It is surprising that the literature review was poorly designed. I think that one paragraph (from line 123 to line 138) is not enough to justify the selection of six hypotheses (H5 - H10). To address this concern, I propose to provide additional support for the literature review. Accordingly, the literature section can benefit from further updated and relevant references. Some examples of papers that may be helpful are listed below:
https://doi.org/10.1080/13548506.2021.1946571
https://doi.org/10.1016/j.dib.2022.108365
https://doi.org/10.1002/casp.2509
Methodology
7. Since the data collection was conducted using an online survey, I am curious whether or not the authors pretested the questionnaire to ensure its comprehensibility.
8. As presented in subsection 3.4, it is clear that authors followed the covariance based Structural Equation Modeling (CB-SEM). Therefore, I would like to inquire about the reason for your decision to choose CB-SEM instead of PLS-SEM.
Results
9. The findings are correctly reported.
Discussion
10. As you have outlined the descriptive statistics in the methodology section (lines 202 -206), I do not see the value of re-exposing them in the discussion section (Lines 302 – 305).
11. I think that 5.1 subsection does not address the theoretical implications, but it provides a discussion of study findings. If not, what is the distinction between a theoretical contribution and theoretical implications?
Minor issues:
12. Avoid using captures for figures. I encourage authors to include the original figures for additional clarity.
13. A number of sentences need to be rephrased because they are copied from other works without giving any credit to the original source. For instance, in lines 170 -171 "Attitudes towards behavior refer to the perception of the individual performing a particular behavior". Copied from https://doi.org/10.1016/j.ijid.2020.07.074. Please reduce the index of similarity, which is currently 36%. If it is about one of your earlier works, kindly avoid self-plagiarism, by reformulating the plagiarized sentences.
14. Some English editing is needed to correct the various grammatical errors present in the manuscript, for instance
· In line 17 "Structure equation model", please correct it; "structural equation modeling".
· In line 18 "latent variable construct", please correct it; "latent variables".
· In line 435 "our sample was collected via an online questionnaire"; We do not collect population; it is correct to state: "Data were collected via an online questionnaire ".
I strongly recommend professional proofreading for the entire manuscript.
Wish you all the best in your research.
Author Response
Dear Authors,
Thank you so much for providing this paper on "Factors affecting preventive behaviors for safety and health at work during the Covid-19 pandemic among Thai construction workers". The study adopted the Ajzen's TPB and the Protection Motivation Theory to explore the determinants of preventive behaviors for safety and health at work during the Covid-19 pandemic. The topic of this study is very interesting. However, the following points are worthy of consideration to improve the manuscript’s comprehensibility.
Introduction
- The study's motivation needs to be clearly demonstrated.
Response: Thank you for your comment. The sentient was added on lines 80-84
Due to construction industry having a significant effect on COVID-19 pandemic, construction workers’ preventive behavior during COVID-19 lack of literature review conducted in Thailand. Thus, it is significant to explore the factors affecting the health and safety-related behavior of construction workers during the COVID-19 pandemic.
Theoretical Research Framework
- As you have mobilized the Ajzen's TPB and the Protection Motivation Theory to develop your theoretical model, I suggest that you provide a description of these two theories before proceeding with the development of the hypotheses.
Response: Thank you for your comment. The description of the two theories were added on lines 98-107.
PMT refers to humans protecting themselves from the perceived health threat when receiving information about the severity of the risk, the perceived vulnerability, and the opportunity to reduce the risk [39, 42-44]. Researchers have used the PMT as a theoretical framework to understand health-related behaviors and to assess humans’ intention to engage in preventive behaviors [45, 47]. People believe that appropriate preventive behavior may reduce the risk of inaction [48]. While TPB refers to the performance of a particular behavior that is determined by the behavioral intention to perform the behavior [54]. It was developed as an instrument by identifying attitudes, self-efficacy, and norms as significant predictors of people's understanding of their interests in deciding to adopt a changed behavior [44, 55].
- Figure 1 illustrates your model, which is generally resulting from a theoretical construction; I suggest placing it at the end of the theoretical research framework section.
Response: Thank you for your comment. I have placed it at the end of the theoretical research framework section.
- The Ajzen's TPB suggests considering the relationships between the three determinants of behavioral intention, i.e., subjective norms, attitudes, and PBC. I am curious about the reason behind your discarding these hypotheses.
Response: Thank you for your comment. Due to our study integration between the Protection Motivation Theory and Ajzen's TPB, Perceived Vulnerability and Perceived Severity interacted among the three variables (i.e., subjective norms, attitudes, and PBC). Thus, the relationships between the three determinants were not necessary.
- I suggest that you revise the research hypothesis phrasing. Please change "a significant influent"by using "a significant influence ".
Response: Thank you for your comment. I have revised a significant influent to “a significant influence”.
- It is surprising that the literature review was poorly designed. I think that one paragraph (from line 123 to line 138) is not enough to justify the selection of six hypotheses (H5 - H10). To address this concern, I propose to provide additional support for the literature review. Accordingly, the literature section can benefit from further updated and relevant references. Some examples of papers that may be helpful are listed below:
https://doi.org/10.1080/13548506.2021.1946571
https://doi.org/10.1016/j.dib.2022.108365
https://doi.org/10.1002/casp.2509
Response: Thank you very much for providing additional support for the literature review. The literature review was added on lines 134-140.
Trifiletti et al., [108] conducted protective behaviors during the Covid-19 pandemic by using the TPB and risk perception, they recommended that intervention and communication strategies to prevent the spreading of COVID-19 should be strongly organized. While Shanka et al. [109] highlighted that awareness of the risk, feelings of responsibility, moral obligations, respectively influenced compliance behavior.
Methodology
- Since the data collection was conducted using an online survey, I am curious whether or not the authors pretested the questionnaire to ensure its comprehensibility.
Response: Thank you for your comment. The sentence was added on lines 237-238
Succeedingly, a pretest questionnaire was conducted to affirm the comprehensibility of the questionnaire for construction workers
- As presented in subsection 3.4, it is clear that authors followed the covariance based Structural Equation Modeling (CB-SEM). Therefore, I would like to inquire about the reason for your decision to choose CB-SEM instead of PLS-SEM.
Response: Thank you for your comment. These two (CB-SEM and PLS-SEM) are widely used methods of SEM. This study studied Factors affecting preventive behaviors, the CB-SEM is better at providing model fit indices. Also, better for factor-based models as the findings of Dash and Pau, (2021).
Results
- The findings are correctly reported.
Response: Thank you for your comment.
Discussion
- As you have outlined the descriptive statistics in the methodology section (lines 202 -206), I do not see the value of re-exposing them in the discussion section (Lines 302 – 305).
Response: Thank you for your comment. We have discussed descriptive statistics in the discussion section on lines 334-346.
Most of construction workers in our study (64.4%) had a lower than high school education. The majority of them were male (62.5%). The primary age group (38%) of them was between 25 to 34 years old. Moreover, 2.8% of the workers were below 15 years old. Thailand, the Labour Protection Act (chapter 5, Clause 45, B.E 2541, 1998) requires that an employer shall not employ a child under fifteen years of age unless they have graduated from a middle school or the competent authority has determined that the work does not cause any harm to the physical and mental health of the young workers [107]. Since the COVID-19 outbreak, the construction camp has been ordered to stop, leading to a post-epidemic construction labor shortage. The young workers joined our study were approved by local authority to work on construction sites as helpers to senior worker in painting, carpentry, and cement works.
- I think that 5.1 subsection does not address the theoretical implications, but it provides a discussion of study findings. If not, what is the distinction between a theoretical contribution and theoretical implications?
Response: Thank you for your comment. In the 5.1 subsections, we have included the study finding and addressed the theoretical implications in every paragraph. For example, In the second paragraph of the 5.1 subsections showed that the first 2 sentences addressed the study finding and the final 2 sentences addressed theoretical implications.
Regarding the knowledge of COVID-19, the SEM revealed that KN has a significant influence on PV (ẞ=0.378; p<0.0001; KN→PV; H3) and PS (ẞ=0.319; p<0.0001; KN→PS; H4). Knowledge of COVID-19 related to understanding the transmission and incubation period of COVID-19 disease, and viral protocol symptoms that could lead to COVID-19 disease, and how hospital treating COVID-19 patients would positively influence perceived vulnerability and severity. These are essential for the preventive spread of the virus. Prasetyo et al. [38] indicated that understanding COVID-19 among Filipinos during reinforced community quarantine significantly influences perceived vulnerability and severity. Thus, if workers receive more accurate COVID-19 information, they could better understand the disease and its effects and symptoms. This could increase their perceived vulnerability and severity.
Minor issues:
- Avoid using captures for figures. I encourage authors to include the original figures for additional clarity.
Response: Thank you for your comment. The captures for figures were removed.
- A number of sentences need to be rephrased because they are copied from other works without giving any credit to the original source. For instance, in lines 170 -171 "Attitudes towards behavior refer to the perception of the individual performing a particular behavior". Copied from https://doi.org/10.1016/j.ijid.2020.07.074. Please reduce the index of similarity, which is currently 36%. If it is about one of your earlier works, kindly avoid self-plagiarism, by reformulating the plagiarized sentences.
Response: Thank you for your comment. I have revised all sentences that may be copied from other works.
- Some English editing is needed to correct the various grammatical errors present in the manuscript, for instance
- In line 17 "Structure equation model", please correct it; "structural equation modeling".
- In line 18 "latent variable construct", please correct it; "latent variables".
- In line 435 "our sample was collected via an online questionnaire"; We do not collect population; it is correct to state: "Data were collected via an online questionnaire".
I strongly recommend professional proofreading for the entire manuscript.
Response: Thank you for your comment. I have revised it accordingly.
Round 2
Reviewer 3 Report (New Reviewer)
The authors have done a good job in responding to highlighted comments. They have produced a complete response that significantly enhances this paper.
This manuscript is a resubmission of an earlier submission. The following is a list of the peer review reports and author responses from that submission.
Round 1
Reviewer 1 Report
This study tries to understand factors that affect the health and safety of construction workers to avoid risky behavior. Overall, the study is original and interesting, the methodology is very thorough, and the conclusions summarize the meaning of the work well. I have a few minor comments:
Introduction (34-180) is well done, and explains how different parameters analyzed match each others: organizational support during epidemics, knowledge about COVID-19, perceived vulnerability, perceived severity, perceived behavioral control, subjective norm, attitude toward behavior, and intention to follow the preventive measure. The introduction is comprehensive of all the information needed to understand the paper, although from an occupational health standpoint, more information about the impact of COVID-19 on workers appears necessary (just as an example, see: Gualano et al. Returning to work and the impact of post COVID-19 condition: a systematic review).
Materials and methods (181-223) are well selected and thoroughly explained, and the use of an online questionnaire survey, which consisted of 9 factors with 43 questions, is quite useful and practical. Structure equation model (SEM) was adopted to analyze the causal relationships 17 among the latent variable construct.
Results (224-270) are well detailed, with integration of graphs and tables, making them easier to understand. Framework appears clear and well explained.
The discussion (391-436) is comprehensive and takes up what has been published in the previous literature on this topic. Considering the impact of SARS-CoV-2 infection and the availability of vaccination, it would be interesting to add a paragraph in the discussion section to introduce and highlight the importance of occupational physicians in implementing workplace campaigns for employees as a prevention measure (just as an example: Gualano et al. Employee participation in workplace vaccination campaigns; a systematic review and meta-analysis), but this is up to the authors as this has not been commonly implemented regarding COVID-19 yet, to the best of my knowledge.
Conclusions (364-381) are complete and encompass all the points discussed above in a clear way.
Author Response
This study tries to understand factors that affect the health and safety of construction workers to avoid risky behavior. Overall, the study is original and interesting, the methodology is very thorough, and the conclusions summarize the meaning of the work well. I have a few minor comments:
Introduction (34-180) is well done, and explains how different parameters analyzed match each others: organizational support during epidemics, knowledge about COVID-19, perceived vulnerability, perceived severity, perceived behavioral control, subjective norm, attitude toward behavior, and intention to follow the preventive measure. The introduction is comprehensive of all the information needed to understand the paper, although from an occupational health standpoint, more information about the impact of COVID-19 on workers appears necessary (just as an example, see: Gualano et al. Returning to work and the impact of post COVID-19 condition: a systematic review).
Response: Thank you for your comment. I have added the impact of COVID-19 on workers on page 2, lines 64-71.
As COVID-19 dissemination is enormously related to individual close contact, it has dramatically impacted construction workers. Gathering among construction workers is requisite for field jobs. Thus, a shift has occurred in occupational risk workers who were faced on account of the spreading of the virus such as limitations in working duties and working hours. Concurrently, workers are also significantly exposed to physiological and psychological stress, fear and anxiety, as well as are susceptible to COVID-19 pandemics.
Materials and methods (181-223) are well selected and thoroughly explained, and the use of an online questionnaire survey, which consisted of 9 factors with 43 questions, is quite useful and practical. Structure equation model (SEM) was adopted to analyze the causal relationships 17 among the latent variable construct.
Response: Thank you for your comment.
Results (224-270) are well detailed, with integration of graphs and tables, making them easier to understand. Framework appears clear and well explained.
Response: Thank you for your comment.
The discussion (391-436) is comprehensive and takes up what has been published in the previous literature on this topic. Considering the impact of SARS-CoV-2 infection and the availability of vaccination, it would be interesting to add a paragraph in the discussion section to introduce and highlight the importance of occupational physicians in implementing workplace campaigns for employees as a prevention measure (just as an example: Gualano et al. Employee participation in workplace vaccination campaigns; a systematic review and meta-analysis), but this is up to the authors as this has not been commonly implemented regarding COVID-19 yet, to the best of my knowledge.
Response: Thank you for your comment, I have added one paragraph in the discussion section about workplace vaccination campaigns on page 16, lines 396-402.
In addition to this study, the SARS-CoV-2 vaccine is significant in minimizing the infection risk and reducing the severity of infection of the current disease. Occupational physicians play a crucial role in implementing vaccination, and vaccination information for workers are needed to provide them with a proper vaccine. Occupational physicians’ workplace vaccine campaigns should be organized to provide thorough vaccination to workers to prevent the spreading of the virus and promote workplace safety.
Conclusions (364-381) are complete and encompass all the points discussed above in a clear way.
Response: Thank you for your comment.
Reviewer 2 Report
This study uses the protection motivation theory and theory of planned behavior to investigate the factors affecting preventive behaviors among Thai construction workers related to safety and health at work during COVID-19.
1- Is there a reference for the claim in Lines 56-59?
2- It is true that there is a lack of literature related to the preventive behavior of construction workers during Covid-19. However, there is plenty of work in the construction industry during Covid-19. The literature review in the manuscript is limited. More studies s required related to construction should be added.
3- Please correct typos e.g., line 92, and line 99. Line 194-196 needs to be simplified. Figure and table numbers in the text need to be corrected e.g Line 187 and Line 236.
4- Line 111-112 about H1 and H2, is its response or support in view of Figure 1.
5- How was such a high response (610) in the online questionnaire survey achieved? In view of the construction industry and construction workforce, the number is surprisingly very high.
6- I have the following concerns because numbers are quite high from the construction industry point of view.
A. Female participation is normally quite less among construction workers (say 10%). The females are 37.5% of the sample. Also, a corresponding correction is needed in Table 1.
B. Construction workers are mostly illiterate in low-developed countries and even in developing countries. Whereas, the mix in the survey indicates 50% with middle high school education.
C. Also response to 'Detected Covid-19 disease'. Yes (66.2%).
Author Response
This study uses the protection motivation theory and theory of planned behavior to investigate the factors affecting preventive behaviors among Thai construction workers related to safety and health at work during COVID-19.
- Is there a reference for the claim in Lines 56-59?
Response: Thank you for your comment, there is no reference claim in lines 56-59.
- This study uses the protection motivation theory and theory of planned behavior to investigate the factors affecting preventive behaviors among Thai construction workers related to safety and health at work during COVID-19.
Response: Thank you for your comment.
- Please correct typos e.g., line 92, and line 99. Line 194-196 needs to be simplified.Figure and table numbers in the text need to be corrected e.g Line 187 and Line 236.
Response: Thank you for your comment, I have revised and corrected all typos on lines 98, 103, and 105. In addition, I have the figure and table number on lines 201
- Line 111-112 about H1 and H2, is its response or support in view of Figure 1.
Response: Thank you for your comment, I have revised the organizational response to organizational support on lines 117-118.
- How was such a high response (610) in the online questionnaire survey achieved? In view of the construction industry and construction workforce, the number is surprisingly very high.
Response: Thank you for your comment. We have provided more information in lines 194-197.
A manager or representatives of each construction project were inquired and contributed an online questionnaire to their workers or co-workers. In addition, Construction workers who are well-reading in Thai were requested. The data collection took place between august 8th and October 3rd.
- I have the following concerns because numbers are quite high from the construction industry point of view.
- A. Female participation is normally quite less among construction workers (say 10%). The females are 37.5% of the sample. Also, a corresponding correction is needed in Table 1.
Response: Thank you for your comment. The entire participants were in divergent projects. In addition, Thailand has many women workforce who work on construction sites (International Labour Organization).
- B. Construction workers are mostly illiterate in low-developed countries and even in developing countries. Whereas, the mix in the survey indicates 50% with middle high school education.
Response: Thank you for your comment. The study focus on workers who work in field job including field engineers, construction inspectors, construction forepersons, general laborers, etc. Also, Construction workers who are well-reading in Thai were requested.
- C. Also response to 'Detected Covid-19 disease'. Yes (66.2%).
Response: Thank you for your comment. Due to construction workers working at work sites, encountering workers is necessary. They may have much chance of getting an infection of the virus from their working environment.
Reviewer 3 Report
The manuscript entitled “Factors Affecting Preventive Behaviors for Safety and Health at Work During the COVID-19 Pandemic Among Thai Construction Workers” is an important work of preventive behavior of construction workers during the COVID-19. However, the manuscript needs to be improved, and much more information about the procedure must be added to allow replication of the study.
1. The introduction section is confusing and there is no logic in the introduction of the subject matter of the paper.
2. Regarding the distribution of the questionnaire, the procedure and design of the study must be described in detail: the form of distribution of the questionnaire (online or offline, if online, what is the distribution medium; what is “a convenience sample” referred to in the article and what is the basis for such a description), the specific time of distribution of the questionnaire (the start and end of the questionnaire collection).
3. The biggest question is whether the authors consider themselves to be exploring Driving Factors or Influencing Factors? The difference between the two is that the former is an internal controllable factor that does not change with the environment, while the latter is an external uncontrollable factor that changes with the environment. If what the authors are exploring is the influencing factor, then I think this is problematic.
I think the optimized model obtained after SEM needs to be added to the article.
Author Response
The manuscript entitled “Factors Affecting Preventive Behaviors for Safety and Health at Work During the COVID-19 Pandemic Among Thai Construction Workers” is an important work of preventive behavior of construction workers during the COVID-19. However, the manuscript needs to be improved, and much more information about the procedure must be added to allow replication of the study.
The introduction section is confusing and there is no logic in the introduction of the subject matter of the paper.
Response: Thank you for your comment. I have added some sentences on lines 64-71.
As COVID-19 dissemination is enormously related to individual close contact, it has dramatically impacted construction workers. Gathering among construction workers is requisite for field jobs. Thus, a shift has occurred in occupational risk workers who were faced on account of the spreading of the virus such as limitations in working duties and working hours. Concurrently, workers are also significantly exposed to physiological and psychological stress, fear and anxiety, as well as are susceptible to COVID-19 pandemics.
- Regarding the distribution of the questionnaire, the procedure and design of the study must be described in detail: the form of distribution of the questionnaire (online or offline, if online, what is the distribution medium; what is “a convenience sample”referred to in the article and what is the basis for such a description), the specific time of distribution of the questionnaire (the start and end of the questionnaire collection).
Response: Thank you for your comment. I have revised accordingly on lines 191-199.
The study focus on workers who work in field job including field engineers, construction inspectors, construction forepersons, general laborers, etc. The entire participants were in divergent projects. A manager or representatives of each construction project were inquired and contributed an online questionnaire to their workers or co-workers. In addition, Construction workers who are well-reading in Thai were requested. The data collection took place between august 8th and October 3rd, 2022. A convenience sample of 610 construction workers was collected to distribute the online questionnaire.
- The biggest question is whether the authors consider themselves to be exploring Driving Factors or Influencing Factors? The difference between the two is that the former is an internal controllable factor that does not change with the environment, while the latter is an external uncontrollable factor that changes with the environment. If what the authors are exploring is the influencing factor, then I think this is problematic.
Response: Thank you for your comment. This study explored factor that affect the health and safety of construction workers to avoid risky behavior. We integrated the Theory of Planned Behavior (TPB) and the protection Motivation Theory (PMT), which provides new insight into construction workers’ preventive behavior during COVID-19 pandemic in Thailand. This study identified Organizational support toward COVID-19 pandemic and Knowledge about COVID-19 as significant variables affecting the preventive behavior of construction workers during the epidemic.
I think the optimized model obtained after SEM needs to be added to the article.
Response: Thank you for your comment. I have added the obtained after SEM in Figure 3, line 264.
Round 2
Reviewer 2 Report
The concerns are not addressed appropriately because now it leads to further concerns about the study. For example, it does not make any distinction between skilled and unskilled labor. The age groups and education level is an important factor for the indicated categories and has not been discussed appropriately.
Reviewer 3 Report
Congratulations!